# Public awareness of acetaminophen and risks of drug induced liver injury: Results of a large outpatient clinic survey

Robert A. Mitchell[1°], Sahaj Rathi[1°], Monica Dahiya[2], Julie Zhu[3], Trana Hussaini[4], Eric M. Yoshida[1]*

**1** Division of Gastroenterology, University of British Columbia, Vancouver, British Columbia, Canada,
**2** Undergraduate School of Medicine, University of British Columbia, Vancouver, British Columbia, Canada,
**3** Division of Digestive Sciences, Dalhousie University, Halifax, Nova Scotia, Canada, **4** Faculty of Pharmaceutical Sciences, University of British Columbia, Vancouver, British Columbia, Canada

° These authors contributed equally to this work.
* eric.yoshida@vch.ca

**Data Availability Statement:** All relevant data are available in OSF (https://osf.io/tmexc/).

**Funding:** There was no funding required or allocated for the study.

## Abstract

Acetaminophen is one of the most commonly consumed analgesics world wide. Generally perceived as a safe medication, it is the most common cause of acute liver failure in the United States with inadvertent hepatotoxicity in half of all cases. We therefore conducted a survey on the public perceptions of acetaminophen in patients attending the outpatient clinic in Vancouver, Canada. Among 928 patients who were asked, 765 completed the survey questionnaire. The majority of respondents were female (59%), Caucasian (61%), and educated beyond the secondary school level (81%). 23% reported using acetaminophen at least once a week. A significant minority were unaware of the potential liver toxicity of acetaminophen (24%), and knowledge of hepatotoxicity did not vary with education status. In terms of the medicinal composition of acetaminophen products, over half of the respondents (58%) did not know that extra strength preparations of acetaminophen contained the same drug but in a different dose. This knowledge was more prevalent among those with higher level of education (49% in graduate school educated respondents), but was still low overall. The knowledge that alcohol use with acetaminophen was more harmful was low (43%), but improved with level of education (P for trend 0.03). Among respondents who consumed alcohol regularly, 21% were consuming over 1.5 grams of acetaminophen at a time. These patients had similar harm perception to liver as patients who consumed lower doses of acetaminophen. Overall, in a large, well-educated cohort of patients, knowledge about the adverse effects of acetaminophen, the additional risks with alcohol and composition of various retailed products was suboptimal. We speculate that consumer ignorance is a significant reason why acetaminophen is a leading cause of acute liver failure.

**Competing interests:** The authors have declared that no competing interests exist.

## Introduction

Acetaminophen (APAP) is an over-the-counter analgesic/antipyretic widely used across the world. APAP has grown increasingly popular in North America since the early 1980s is now the most commonly used over-the-counter analgesic/antipyretic in the United States with over 28 billion doses distributed annually [1]. APAP is available over the counter (OTC) in several dosage formulations including 325 mg, 500 mg and 650 mg tablets/capsules and it is an ingredient of other combination products including prescription analgesics (in combination with opiate products) as well as other OTC "cold" products such as cough syrups. The retailing of APAP products is also ubiquitous. APAP products are sold in community pharmacies but also convenience stores, grocery stores, gas stations, hotel lobby gift shops etc with no restrictions on the quantity that can be purchased.

Despite APAP's immense popularity and ubiquity as an analgesic agent, it is also a dose-related toxin[2]. APAP is generally safe when used at its manufacturer recommended dose of less than 4 grams per day[3]. Unfortunately, the therapeutic index of APAP is relatively narrow, and significant adverse effects commonly result from doses exceeding 10 grams in 24 hours[4]. The most common and feared adverse event of APAP overdose is hepatotoxicity and acute liver failure. In the 1990s APAP toxicity was thought to account for roughly 20% of all acute liver failure diagnoses[5]. This has concerningly increased over the past two decades, with recent data from a US longitudinal registry cohort study of acute liver failure (ALF) (defined as liver failure with the presence of hepatic encephalopathy, coagulopathy and no previous liver disease) reporting that APAP hepatotoxicity is the single most common cause of ALF in the United States occurring in 46% of reported cases with over half the cases of APAP being inadvertent[6]. In the UK and Europe, APAP has been implicated in 40–70% cases of acute liver failure[7]. ALF from APAP continues to carry a significant risk of mortality with transplant free survival between 2006 to 2013 reported as 75.6% in the United States[6].

Although intentional self-harm contributes to APAP toxicity, it has been consistently documented that accidental toxicity related to therapeutic use of APAP, commonly known as APAP "therapeutic misadventure", is an extremely common mechanism of toxicity, especially when combined with alcohol[8]. Patients commonly underestimate the toxicity of APAP[9] especially in the setting of alcohol consumption, or fail to recognize when they are inadvertently taking more than one APAP-containing product[10]. It is unclear if this effect is less pronounced in patients with known liver disease who should theoretically be more aware of the hepatotoxic effects of APAP.

There is, therefore, a clear need for better patient education and possible regulatory action with respect to APAP. The current study aims to evaluate patient understanding of the hepatotoxic effects of APAP. This knowledge can be used as a bridge to improved patient education and possible future regulatory action around this issue.

## Aims

The aim of this study was to assess patient understanding of APAP intake in a general gastroenterology outpatient clinic population. We specifically set out to

- Quantify intake of regular and extra-strength APAP

- Evaluate concurrent alcohol use patterns with APAP

- Explore patient understanding of hepatoxic effects of APAP alone and combined with alcohol

- Determine if patients understand the difference between the regular strength products and the extra strength products

## Methods

### Study population

We performed a clinic-administered patient survey during patient visits to a general gastroenterology clinic. Between October 2018 and March 2019, unselected consecutive patients aged $\geq$ 18 years were administered the survey for optional completion while they wait for their appointment to begin. Patients were given a copy of the study questionnaire (S1 Appendix) to self-complete either before or after their appointment. An explanatory letter was given to each patient at the time of the survey, and return of the survey was accepted as an implied consent (explained in the explanatory letter). Upon completion and return of the survey, all patients were given an educational information sheet on acetaminophen and liver injury.

### Exclusion criteria

- Unable to use a writing utensil without assistance

- Unable to speak or understand English–due to staff language limitations

**Table 1. Demographics of respondents.**

|  | N | % |
|---|---|---|
| **Total** | 765 | |
| **Males** | 312 | 41 |
| **Ethnicity** | | |
| • Caucasian | 470 | 61% |
| • East/South East Asian | 190 | 25% |
| • South Asian | 42 | 5% |
| • Hispanic | 17 | 2% |
| • Middle eastern | 17 | 2% |
| • Indigenous | 9 | 1% |
| • Afro-Carribean | 5 | 1% |
| • Other | 10 | 1% |
| **Country of birth** | | |
| • Canada | 422 | 55% |
| • China | 71 | 9% |
| • India | 21 | 3% |
| • USA | 20 | 3% |
| • Other | 231 | 30% |
| **Highest Education** | | |
| • Grade School | 19 | 2% |
| • High School | 122 | 16% |
| • College/University | 459 | 60% |
| • Graduate school | 163 | 21% |
| **Alcohol consumption** | | |
| • Never | 295 | 39% |
| • Less than one drink/week | 202 | 26% |
| • 2–3 drinks/week | 105 | 14% |
| • 4–7 drinks/week | 77 | 10% |
| • 8–14 drinks/week | 32 | 4% |
| • > 14 drinks/week or binge drinking | 22 | 3% |
| • No response | 31 | 4% |

The University of British Columbia Behavioural Research Ethics Board approved this study (approval certificate number H18-01933). This approval was obtained prior to administering the survey. No identifying patient data was collected.

Sample size calculation: As this was a descriptive patient awareness survey with minimal pre-existing information about prevalent level of information in the population with respect to the administered questionnaire, a sample size calculation was not done.

## Data analysis

Data will be expressed as percentages for categorical variables and mean values for continuous variables. Chi-square tests or Fischer exact test were used to assess study results for categorical variables. P-values (2-tailed) < 0.05 were interpreted as significant.

## Results

### Patient demographics

A total of 928 patients attending the gastroenterology clinic at the Vancouver General Hospital were offered the APAP Perception Questionnaire. Of those, 765 completed the survey, and were considered for analysis. A majority of the respondents were females (59%), Caucasian (61%), and born in Canada (55%). The demographics of study population are described in Table 1. Over 80% of the respondents had college/university or higher education.

### APAP consumption

About a quarter of respondents (23%) reported using APAP at least once a week in the past year. Only two respondents admitted consuming over 4 grams of APAP in a day. Similar patterns of both regular and extra strength APAP use were seen among ethnic groups and education levels in terms of frequency as well as dosing. (Fig 1)

### APAP perceptions

A majority of the respondents (76%) were familiar with liver toxicity of APAP. Over half of the patients perceived APAP was harmful to other organs as well (Table 2). Only 6% patients perceived APAP as completely safe.

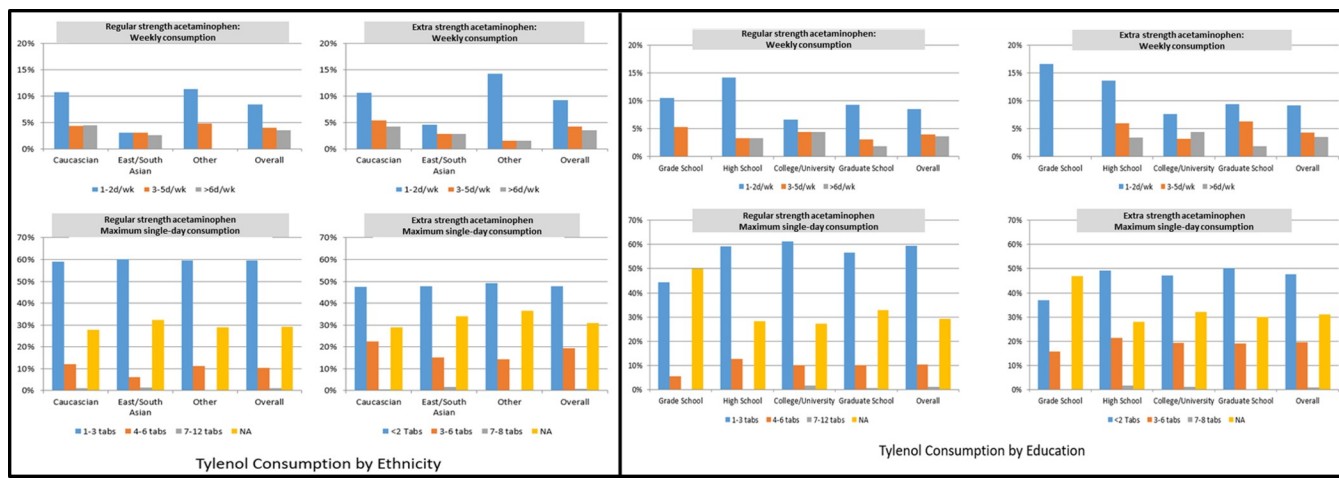

**Fig 1. Profile of acetaminophen consumption across ethnic and education classes.**

### Regular acetaminophen (325 mg formulation) vs extra strength acetaminophen (500 mg, 650 mg formulations)

Almost a third of patients felt that extra strength acetaminophen products were APAP combined with a different drug that provided additional pain-relief or was a different product altogether. The knowledge that it was the same medication (ie. APAP) increased with education level, and the trend was statistically significant (P<0.001) (Fig 2)

### APAP and alcohol

The perception that consumption of alcohol in combination with APAP was more harmful was significantly different across education classes (Fig 3) (P for trend 0.03) Among patients with regular alcohol intake more than 7 drinks/week, we considered the threshold of high dose APAP consumption to be >1.5 grams at a time. There was no difference in frequency of high dose APAP consumption between regular alcohol consumers and those who drank alcohol less frequently(21% vs 32%, p = 0.11). Respondents consuming high dose APAP had similar perception of harm as patients consuming<1.5 grams (Perception of harm to liver—75% vs 81%, P = 0.39).

## Discussion

This study assesses patient perceptions about APAP in a large number of out-patient respondents across a diverse educational and ethnic spectrum. The majority of our respondents were women, Caucasian, and well-educated (60% beyond high school and 20% beyond the undergraduate university level). We confirm that regular APAP use is common in patients attending outpatient clinics. Although most patients (76%) are well aware of liver toxicity of APAP, a significant minority are not. This number was similar to what has previously been reported.[9] However, there is also a fair amount of misinformation of toxicity in other organs. Over half of the patients feel APAP is harmful to the stomach, intestines, heart, or brain. Whether this was significantly different among education classes was difficult to determine due to low numbers in each category.

We also realized that there was a gap in knowledge among patients about extra-strength APAP. Nearly 60% patients were not aware that it is the same drug as regular APAP but simply formulated with a higher dosage. This knowledge was more prevalent in respondents with higher level of education. However, even among respondents having attended university or graduate school, less than 50% were aware of the true nature of the drug. This would imply that consumers of APAP are not aware of how much of the drug they are actually ingesting. The potential for inadvertent hepatotoxicity would appear to be high as taking two to three

**Table 2. Perceptions of acetaminophen toxicity among respondents.**

|  | N | % |
|---|---|---|
| Regular APAP Use (>1/week) | 174 | 23% |
| Harm perceptions |  |  |
| • No harm | 50 | 6% |
| • Heart | 76 | 10% |
| • Liver | 584 | 76% |
| • Pancreas | 148 | 19% |
| • Stomach | 368 | 48% |
| • Brain | 63 | 8% |
| • Intestine and Colon | 201 | 26% |

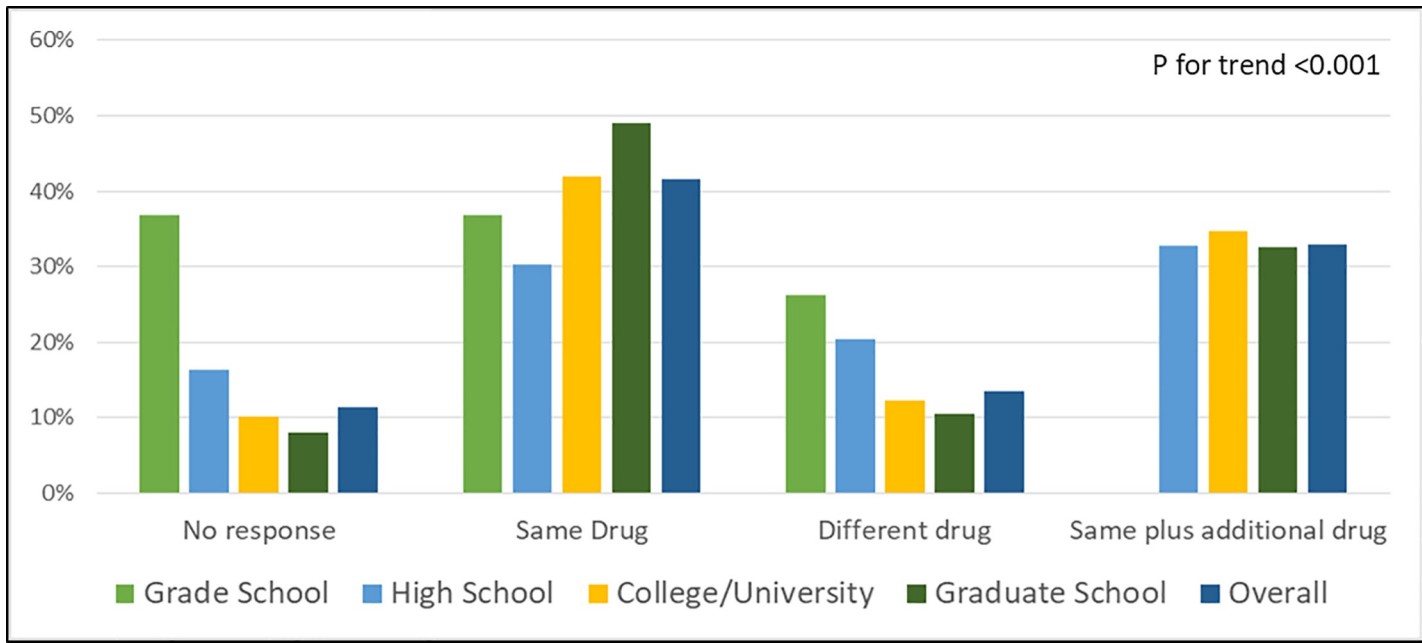

**Fig 2. Prevalence of awareness about difference between regular strength and extra-strength acetaminophen preparations.**

tablets four times daily would deliver a dosage below the total recommended daily dose of 4 g per day in the cases of regular strength preparations (ie. 325 mg tablet/capsule), while taking the exact same number of pills in the form of extra-strength preparations (500 to 650 mg tablets/capsules) would results in dosages (6-8g per day) far exceeding the total recommended daily dose of APAP. It is perhaps not surprising that APAP drug induced liver injury (DILI) is the leading cause of serious ALF in the United States and that most is unintentional, inadvertent DILI[6]. A question for governmental regulators of consumer products should be whether the extra strength formulations of APAP should be marketed at all to eliminate the confusion regarding the total daily dosage of APAP consumed. Since all the different preparations of

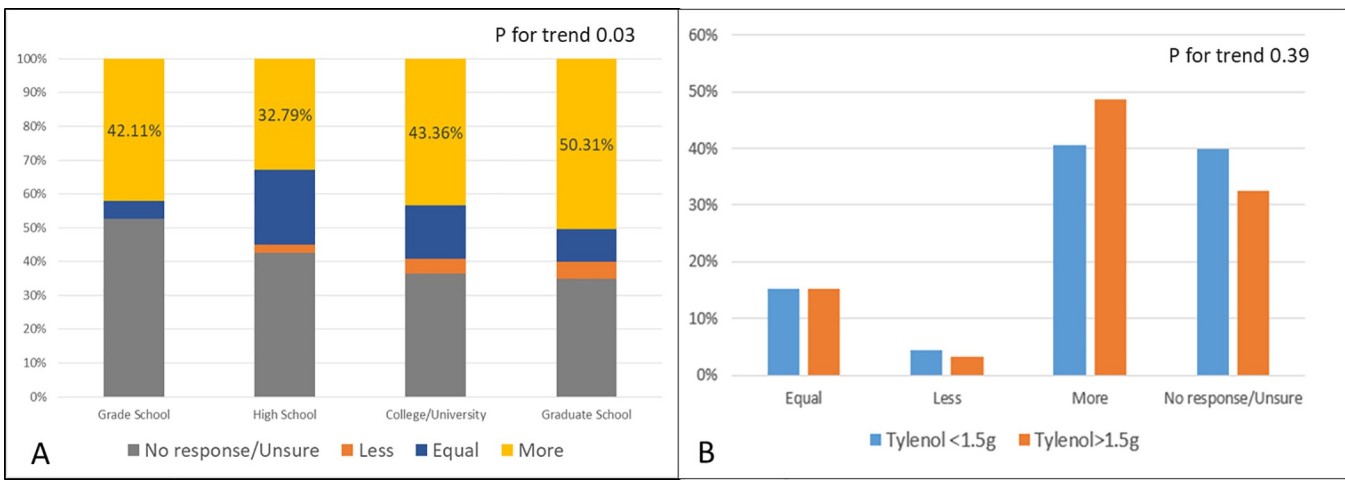

**Fig 3.** Perception of toxicity of acetaminophen when used in combination with alcohol across education classes(A), and comparing respondents consuming more than 1.5 grams of acetaminophen in a day(B).

APAP essentially contain the same product with varying strength, those who feel the need to take more acetaminophen could simply consume an extra 325 mg tablet/capsule. Limiting the quantities that can be purchased at any given time would also appear to be a prudent consumer safety measure.

Nearly 60% of our respondents were unaware of additional toxicity of APAP when used in combination with alcohol. While there was a difference in this knowledge across educational status, over half of the respondents educated beyond university level were unaware of this fact. The overall lack of awareness of APAP dosage strength in a popular over the counter medication combined with a lack of awareness of potential hepatotoxicity, in a significant minority, suggests that the potential risk of inadvertent APAP hepatotoxicity in the general population is currently very high. Our findings would also provide an easy explanation as to why APAP is overwhelmingly the leading cause of ALF in America[6].

The usually accepted threshold for APAP hepatotoxicity is 4 grams/day. However, among patients with liver disease and/or chronic alcohol intake, the safe limit is thought to be 2 grams a day. The threshold might be even lower for those taking APAP chronically. We conservatively used the threshold of 1.5 grams for risk estimation to account for under-reporting and recall bias. Among our respondents who consumed alcohol regularly, nearly 20% admitted having taken more than 1.5 grams of APAP in a day. A majority of these respondents were well aware of liver toxicity of APAP, and that the combination of the two was more harmful.

While only 2 of 765 respondents admitted to having consumed more than 4 grams of APAP in a day, nearly a fifth of chronic alcohol consumers admitted to have taken APAP beyond the threshold of toxicity. In our relatively well educated cohort, this reflects a lack of awareness about the threshold of toxicity of APAP. We suggest that public education regarding the risks of APAP hepatoxicity, especially in the context of alcohol use, is necessary.

The main strengths of this study are the large number of respondents, and the diverse ethnic spectrum. In fact, to the best of our knowledge, this is the largest survey of its kind that has ever been conducted in Canada. Moreover, a majority of the participants were educated beyond secondary education level. The prevalence of misinformation in this population group likely represents the lack of easy access to information rather than simply patient apathy.

The major limitations of our study include the single centre, single specialty office respondent cohort. Although this may limit the generalizability of the study, we suspect that similar results would be found throughout North America and, given the high degree of education of the surveyed cohort, similar surveys in less educated cohorts are likely to reveal even less awareness. Unfortunately, we could not use a validated questionnaire due to the paucity of research in this field. However, the large number of our respondents did act as a partial internal validation for our questionnaire.

## Conclusions

A quarter of patients attending the gastroenterology clinic were using APAP regularly. Most patients were aware of liver toxicity of APAP, however, almost a quarter of those surveyed, were unaware. A majority of patients were unaware of the pharmaceutical content of different products of APAP. Most patients were aware that APAP had additional toxicity when combined with alcohol use, however, a substantial proportion of patients chronically consuming alcohol were still taking unsafe amount of APAP. This indicated a possible dichotomy between knowledge and personal practice that may stem from a lack of direct knowledge on the risks associated with APAP use. Our study strongly suggests that better public awareness about APAP in general, and hepatotoxicity in particular, is needed. Government consumer regulation regarding the availability of the higher dosage formulations of APAP, as well as the

quantities sold, may also be necessary to protect the public from acetaminophen related hepatotoxicity which commonly results in health care utilization and unfortunate personal tragedies.

## Supporting information

**S1 Appendix. Study survey.**
(DOCX)

**S2 Appendix. Patient handout on acetaminophen and liver toxicity.**
(DOCX)

## Acknowledgments

We wish to thank Dr. Baljinder Salh for his administrative assistance with this study.

## Author Contributions

**Conceptualization:** Trana Hussaini, Eric M. Yoshida.

**Data curation:** Robert A. Mitchell, Monica Dahiya, Julie Zhu.

**Formal analysis:** Sahaj Rathi.

**Investigation:** Robert A. Mitchell, Julie Zhu.

**Methodology:** Monica Dahiya, Julie Zhu, Eric M. Yoshida.

**Project administration:** Monica Dahiya, Julie Zhu, Eric M. Yoshida.

**Resources:** Monica Dahiya, Trana Hussaini, Eric M. Yoshida.

**Supervision:** Trana Hussaini, Eric M. Yoshida.

**Validation:** Sahaj Rathi, Trana Hussaini, Eric M. Yoshida.

**Visualization:** Sahaj Rathi.

**Writing – original draft:** Sahaj Rathi.

**Writing – review & editing:** Robert A. Mitchell, Trana Hussaini, Eric M. Yoshida.

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
