## [Decision Letter · Decision Letter 0]

24 Jan 2020

PONE-D-19-30395

PUBLIC AWARENESS OF ACETAMINOPHEN AND RISKS OF DRUG INDUCED LIVER INJURY: RESULTS OF A LARGE OUTPATIENT CLINIC SURVEY

PLOS ONE

Dear Dr. Yoshida,

Thank you for submitting your manuscript to PLOS ONE. After careful consideration, we feel that it has merit but does not fully meet PLOS ONE’s publication criteria as it currently stands. Therefore, we invite you to submit a revised version of the manuscript that addresses the points raised during the review process.

As you can see, both reviewers appreciated your paper, but provided suggestions that should be considered during the revision process.

We would appreciate receiving your revised manuscript by Mar 09 2020 11:59PM. To enhance the reproducibility of your results, we recommend that if applicable you deposit your laboratory protocols in protocols.io, where a protocol can be assigned its own identifier (DOI) such that it can be cited independently in the future. For instructions see: http://journals.plos.org/plosone/s/submission-guidelines#loc-laboratory-protocols

We look forward to receiving your revised manuscript.

Kind regards,

Pavel Strnad

Academic Editor

PLOS ONE

2. Please provide a sample size and power calculation in the Methods, or discuss the reasons for not performing one before study initiation.

3. In your Methods section, please provide the date range (month and year) during which participants were invited to complete the questionnaire.

4. Thank you for including your ethics statement: "University of British Columbia REB approval number H18-01933".

a) Please amend your current ethics statement to confirm that your named institutional review board or ethics committee specifically approved this study.

a)    Please provide an amended Funding Statement that declares *all* the funding or sources of support received during this specific study (whether external or internal to your organization) as detailed online in our guide for authors at http://journals.plos.org/plosone/s/submit-now.  

b)    Please state what role the funders took in the study.  If any authors received a salary from any of your funders, please state which authors and which funder. If the funders had no role, please state: "The funders had no role in study design, data collection and analysis, decision to publish, or preparation of the manuscript."

6. We note that you have indicated that data from this study are available upon request. PLOS only allows data to be available upon request if there are legal or ethical restrictions on sharing data publicly. For information on unacceptable data access restrictions, please see http://journals.plos.org/plosone/s/data-availability#loc-unacceptable-data-access-restrictions.

7. PLOS requires an ORCID iD for the corresponding author in Editorial Manager on papers submitted after December 6th, 2016. Please ensure that you have an ORCID iD and that it is validated in Editorial Manager. To do this, go to ‘Update my Information’ (in the upper left-hand corner of the main menu), and click on the Fetch/Validate link next to the ORCID field. This will take you to the ORCID site and allow you to create a new iD or authenticate a pre-existing iD in Editorial Manager. Please see the following video for instructions on linking an ORCID iD to your Editorial Manager account: https://www.youtube.com/watch?v=_xcclfuvtxQ

8. We note that you have included the phrase “data not shown” in your manuscript. Unfortunately, this does not meet our data sharing requirements. PLOS does not permit references to inaccessible data. We require that authors provide all relevant data within the paper, Supporting Information files, or in an acceptable, public repository. Please add a citation to support this phrase or upload the data that corresponds with these findings to a stable repository (such as Figshare or Dryad) and provide and URLs, DOIs, or accession numbers that may be used to access these data. Or, if the data are not a core part of the research being presented in your study, we ask that you remove the phrase that refers to these data.

Reviewers' comments:

Reviewer's Responses to Questions

**Comments to the Author**

1. Is the manuscript technically sound, and do the data support the conclusions?

Reviewer #1: Yes

Reviewer #2: Yes

2. Has the statistical analysis been performed appropriately and rigorously? 

Reviewer #1: Yes

Reviewer #2: Yes

3. Have the authors made all data underlying the findings in their manuscript fully available?

Reviewer #1: Yes

Reviewer #2: No

4. Is the manuscript presented in an intelligible fashion and written in standard English?

Reviewer #1: Yes

Reviewer #2: Yes

5. Review Comments to the Author

Reviewer #1: I read with great pleasure the manuscript entitled "PUBLIC AWARENESS OF ACETAMINOPHEN AND RISKS OF DRUG INDUCED LIVER INJURY: RESULTS OF A LARGE OUTPATIENT CLINIC SURVEY". Except praise, I have no further comments, and strongly support publishing of this manuscript.

Reviewer #2: The topic is extremely important because of the flu season and the rate of liver damage I see due to the incorrect use of drugs. The authors speculate that consumer ignorance is a significant reason why acetaminophen is a leading cause of acute liver failure. I think, it may be a long way. In terms of the medicinal composition of acetaminophen products, over half of the respondents (58%) did not know that extra strength preparations of acetaminophen contained the same drug but in a different dose, but 81% (!) were educated beyond the secondary school level. My concern is not the education level, but the English/French proficiency, because only 55% were born in Canada! The authors should better discussed on this point than speculate about the "ignorance level" in my opinion. Otherwise stats are fine. I would also suggest compare the results of this survey with similar (not equal) surveys potentially performed in the UK, Australia, US, and France. The authors may contact the respective Medical Councils to receive anonymous data.

6. PLOS authors have the option to publish the peer review history of their article (what does this mean?). If published, this will include your full peer review and any attached files.

Reviewer #1: No

Reviewer #2: Yes: Consolato SERGI

---

## [Author Response · Author response to Decision Letter 0]

27 Jan 2020

Reviewer #1: I read with great pleasure the manuscript entitled "PUBLIC AWARENESS OF ACETAMINOPHEN AND RISKS OF DRUG INDUCED LIVER INJURY: RESULTS OF A LARGE OUTPATIENT CLINIC SURVEY". Except praise, I have no further comments, and strongly support publishing of this manuscript.

We thank the reviewer for their appreciation.

Reviewer #2: The topic is extremely important because of the flu season and the rate of liver damage I see due to the incorrect use of drugs. The authors speculate that consumer ignorance is a significant reason why acetaminophen is a leading cause of acute liver failure. I think, it may be a long way. In terms of the medicinal composition of acetaminophen products, over half of the respondents (58%) did not know that extra strength preparations of acetaminophen contained the same drug but in a different dose, but 81% (!) were educated beyond the secondary school level. My concern is not the education level, but the English/French proficiency, because only 55% were born in Canada! The authors should better discussed on this point than speculate about the "ignorance level" in my opinion. 

We appreciate the concerns put forth by the reviewer. Interestingly, this high proportion of respondents being born outside Canada simply reflects the shifting demographics of metro Vancouver. Our survey specifically excluded those who were not proficient in English, and our protocol that was approved by ethics, accepted this as an exclusion criteria. Moreover, French isn’t commonly spoken in Vancouver. In fact, Mandarin, Cantonese and Punjabi are more common.

Otherwise stats are fine. I would also suggest compare the results of this survey with similar (not equal) surveys potentially performed in the UK, Australia, US, and France. The authors may contact the respective Medical Councils to receive anonymous data.

We agree that similar surveys from other countries would be valuable. However, to compare results we would need a survey already conducted which is similar to ours. Unfortunately, we could not find any mention of a comparable survey anywhere in literature. Any surveys we could find which looked at similar variables have been covered in introduction and discussion. We hope this would be acceptable to the reviewers and editors to go forward and publish this paper. Hopefully, our paper may be used as a scaffolding to generate similar surveys in other places.

---

## [Decision Letter · Decision Letter 1]

30 Jan 2020

PUBLIC AWARENESS OF ACETAMINOPHEN AND RISKS OF DRUG INDUCED LIVER INJURY: RESULTS OF A LARGE OUTPATIENT CLINIC SURVEY

PONE-D-19-30395R1

Dear Dr. Yoshida,

We are pleased to inform you that your manuscript has been judged scientifically suitable for publication and will be formally accepted for publication once it complies with all outstanding technical requirements.

With kind regards,

Pavel Strnad

Academic Editor

PLOS ONE

Additional Editor Comments (optional):

Reviewers' comments:

Reviewer's Responses to Questions

**Comments to the Author**

1. If the authors have adequately addressed your comments raised in a previous round of review and you feel that this manuscript is now acceptable for publication, you may indicate that here to bypass the “Comments to the Author” section, enter your conflict of interest statement in the “Confidential to Editor” section, and submit your "Accept" recommendation.

Reviewer #2: All comments have been addressed

2. Is the manuscript technically sound, and do the data support the conclusions?

Reviewer #2: Yes

3. Has the statistical analysis been performed appropriately and rigorously? 

Reviewer #2: Yes

4. Have the authors made all data underlying the findings in their manuscript fully available?

Reviewer #2: Yes

5. Is the manuscript presented in an intelligible fashion and written in standard English?

Reviewer #2: Yes

6. Review Comments to the Author

Reviewer #2: The authors addressed the comments and suggestions of the reviewers.

7. PLOS authors have the option to publish the peer review history of their article (what does this mean?). If published, this will include your full peer review and any attached files.

Reviewer #2: Yes: Consolato Sergi

---

## [Editor Report · Acceptance letter]

6 Feb 2020

PONE-D-19-30395R1 

Public awareness of acetaminophen and risks of drug induced liver injury: results of a large outpatient clinic survey 

Dear Dr. Yoshida:

I am pleased to inform you that your manuscript has been deemed suitable for publication in PLOS ONE. Congratulations! Your manuscript is now with our production department. 

With kind regards,

on behalf of

Dr. Pavel Strnad 

Academic Editor

PLOS ONE